# Identification of the Porcine Vascular Endothelial Cell-Specific Promoter ESAM1.0 Using Transcriptome Analysis

**DOI:** 10.3390/genes14101928

**Published:** 2023-10-11

**Authors:** Sang Eun Kim, Wu-Sheng Sun, Miae Oh, Seunghoon Lee, Jin-Gu No, Haesun Lee, Poongyeon Lee, Keon Bong Oh

**Affiliations:** 1Animal Biotechnology Division, National Institute of Animal Science, Rural Development Administration, Jeonju-si 55365, Jeollabuk-do, Republic of Korea; sesesese@korea.kr (S.E.K.); sunwsh@jlau.edu.cn (W.-S.S.); miae9550@gmail.com (M.O.); darkcherub@korea.kr (S.L.); shrkftm@korea.kr (J.-G.N.); leehs1498@korea.kr (H.L.); pylee@korea.kr (P.L.); 2College of Veterinary Medicine, Jilin Agricultural University, Changchun 130118, China

**Keywords:** ESAM, porcine promoter, porcine vascular endothelial cells, xenotransplantation

## Abstract

The vascular endothelium of xenografted pig organs represents the initial site of rejection after exposure to recipient immune cells. In this study, we aimed to develop a promoter specific to porcine vascular endothelial cells as a step toward overcoming xenograft rejection. Transcriptome analysis was performed on porcine aortic endothelial cells (PAECs), ear skin fibroblasts isolated from *GGTA* knockout (GTKO) pigs, and the porcine renal epithelial cell line pk-15. RNA sequencing confirmed 243 differentially expressed genes with expression changes of more than 10-fold among the three cell types. Employing the Human Protein Atlas database as a reference, we identified 34 genes exclusive to GTKO PAECs. The endothelial cell-specific adhesion molecule (*ESAM*) was selected via qPCR validation and showed high endothelial cell specificity and stable expression across tissues. We selected 1.0 kb upstream sequences of the translation start site of the gene as the promoter ESAM1.0. A luciferase assay revealed that ESAM1.0 promoter transcriptional activity was significant in PAECs, leading to a 2.8-fold higher level of expression than that of the porcine intercellular adhesion molecule 2 (*ICAM2*) promoter, which is frequently used to target endothelial cells in transgenic pigs. Consequently, ESAM1.0 will enable the generation of genetically modified pigs with endothelium-specific target genes to reduce xenograft rejection.

## 1. Introduction

Xenotransplantation, the transplantation of organs or cells from one species to a different species, can potentially address organ shortage challenges [1,2,3]. Owing to their anatomical and physiological analogy to humans, pigs have been deemed the most suitable species for this purpose [4]. However, immunological incompatibilities between humans and pigs pose significant obstacles that must be surmounted for successful xenotransplantation. Unlike humans and primates, pigs express the galactose-α-1,3-galactose (α-Gal) antigen, which induces hyperacute rejection [5,6]. This major xenoantigen was eliminated in pigs by knocking out the α-1,3-galactosyltransferase (*GGTA*) gene, demonstrating that genetic modification provides clues to overcoming xenotransplantation obstacles [7,8,9]. Considerable research has been conducted to generate transgenic pigs using gene editing technology, allowing precise and specific endogenous gene ablation as well as foreign gene integration [4,10,11,12].

Endothelial cells, which line the blood vessels of transplanted organs, express multiple xenoantigens [13]. These are the initial cells to communicate with the immune system of the recipient [14]. In the early phases of xenorejection, porcine endothelial cells are activated and damaged. By driving transgene expression, specifically in endothelial cells, it is possible to incorporate genetic modifications that reduce the rejection processes and improve graft survival. The promoters of several genes have been identified and studied for their specificity to porcine endothelial cells, with porcine intercellular adhesion molecule 2 (*ICAM2*) being one example. The ICAM2 promoter has been frequently used for the precise expression of anti-coagulant and anti-inflammatory proteins [12,15,16]. The Tie1 promoter, another porcine promoter, has also been developed to drive selective gene expression in the endothelial cells of transgenic pigs [17]. Unfortunately, the expression levels of transgenes driven by ICAM2 and Tie1 promoters in transgenic pigs are inadequate [15,18].

In this study, we aimed to design a promoter specific to porcine vascular endothelial cells. First, transcriptome analysis was performed on porcine aortic endothelial cells (PAECs) to pinpoint specific genes. Comparison with the human endothelial cell-specific gene list yielded potential candidate genes. Subsequently, endothelial cell-specific adhesion molecule (*ESAM*) promoter-driven transcriptional activity was confirmed. We finally developed the ESAM1.0 promoter, which controls endothelial cell-specific gene expression. This promoter might serve as a novel tool, providing precise control over gene expression for this essential cell population, which will aid in advancing xenotransplantation.

## 2. Materials and Methods

### 2.1. Cells and Transfection

PAECs and porcine ear fibroblasts (PEFs) were isolated from a 1-month-old GGTA knockout (GTKO) pig [19,20]. The PAECs were then grown in endothelial cell growth medium-2 (Lonza, Basel, Switzerland). The PEFs were cultured in Dulbecco’s modified Eagle’s medium (DMEM, Gibco, Grand Island, NY, USA) supplemented with 10% fetal bovine serum (Gibco), antibiotic-antimycotic (Gibco), non-essential amino acids (Gibco), low serum growth supplement (Gibco), and 2-mercaptoethanol (Gibco). The pk-15 cells were cultured as previously described [21]. Briefly, the cells were maintained in DMEM (Gibco), supplemented with 10% fetal bovine serum (Gibco) and antibiotic-antimycotic (Gibco). The cells used in this study were incubated at 37 °C in a humidified atmosphere containing 5% CO_2_. Transfections for PAECs and PEFs were performed using the Amaxa basic nucleofector (Lonza) according to the manufacturer’s instructions. The pk-15 cells were transfected using a Lipofectamine 3000 Kit (Invitrogen, CA, USA), according to the manufacturer’s protocol. pGL3 vectors were used for the luciferase assays.

### 2.2. RNA Sequencing

Total RNA was extracted from the GTKO PAECs, GTKO PEFs, and pk-15 cells using the Trizol method. The RNA sequencing was performed by TNT Research, Ltd. (Sehong-si, Republic of Korea). The sequencer’s raw reads were preprocessed to remove the adapter and low-quality sequences and were then aligned using HISAT v2.1.0 to the Sus scrofa (Sscrofa11.1) genome [22]. A global, whole-genome index and tens of thousands of small local indexes are utilized by HISAT for alignment. The indexes for these two varieties were built using the same BWT (Burrows–Wheeler transform) and graph FM index (GFM) as for Bowtie2. The reference genome sequence of Sus scrofa (Sscrofa 11.1) and the annotation information were obtained from the NCBI Genome Assembly browser “https://www.ncbi.nlm.nih.gov (assessed on 16 February 2021)”, using StringTie to assemble the transcripts and estimate their abundance [23,24]. Subsequently, StringTie v2.1.3b was utilized to construct aligned reads into the transcripts and estimate their abundance. It provides relative abundance estimates in the form of read count values for each transcript and gene expressed in each sample. StringTie v2.1.3b was also used to assemble known transcripts, novel transcripts, and transcripts with alternative splicing. Read count or FPKM value (fragments per kilobase of exon per million fragments mapping) was used to determine the abundance of transcript and gene expression per sample. The expression profiles are applied to DEG (differentially expressed gene) analysis.

### 2.3. DEG Analysis

The edgeR system was used to perform statistical analyses of differential gene expression with raw counts as input [25]. In the QC step, genes with non-zero counts in at least one group of replicates were selected. RLE normalization with a filtered data set was applied to correct the variations in library sizes among samples. The statistical significance of differential gene expression was determined using nbinomLRT with full models of “Subject+Period group” and a reduced model of “Subject”. The fold change and *p*-value were obtained from the nbinomLRT output. The Benjamini–Hochberg algorithm was applied to all *p*-values to control the false discovery rate (FDR). The list of significant genes was filtered by |fold change| ≥ 2 and a raw *p*-value of less than 0.05 (Appendix A). With these parameters (distance metric = Euclidean distance, linkage method = complete), hierarchical clustering was conducted on rlog-transformed values for significant genes. Using gProfiler “https://biit.cs.ut.ee/gprofiler/orth (assessed on 16 February 2021” and the Gene Ontology (GO) database, gene-enrichment analysis and functional annotation analysis for significant genes were performed [26]. The adjusted *p*-values reported by gProfiler were calculated using a one-sided hypergeometric test and the Benjamini–Hochberg method.

### 2.4. Real-Time Quantitative PCR

Total RNA was extracted from the cells and tissues using an RNeasy Mini Kit (Qiagen, Hilden, Germany). The tissues of wild-type miniature pigs were kindly provided by Dr. Hwang of the Korea Institute of Toxicology. cDNA was synthesized using a Superscript IV First-Strand Synthesis Kit (Invitrogen, Waltham, MA, USA). Real-time quantitative PCR (qPCR) was performed in triplicate using the Power SYBR Green PCR Master Mix (Applied Biosystems, Foster City, CA, USA) and the StepOne Real-Time PCR system (Applied Biosystems). The sequences of the primers used in the assays are presented in Table 1. In the porcine cells, ESAM, platelet endothelium activation receptor 1 (PEAR1), and ICAM2 expression values were normalized to the expression of β-actin (ACTB). The gene expression in tissues was normalized using ACTB and porcine 18S ribosomal RNA. Comparative C_T_ (ΔΔC_T_) quantitation was used to evaluate the relative expression of the genes. ΔΔC_T_ values were determined by analyzing the data using the StepOne software v2.3.

### 2.5. Promoter Isolation and Vector Construction

We screened porcine *ESAM* sequences (NC_010451.4, chr9: 51,983,019–51,992,415), which spanned seven exons, along with approximately 4.9 kb of upstream sequences obtained from the NCBI Sscrofa11.1 database. CpG islands were predicted using Methprimer [27]. ESAM1.0 and ESAM1.5 promoters were designed with NheI and HindIII restriction enzyme sites. For the luciferase assay, putative ESAM and ICAM2 promoters were subcloned into the pGL3-basic luciferase reporter vector (Promega, Madison, WI, USA).

### 2.6. Luciferase Assay

Promoter activity was quantified using a dual-luciferase reporter assay kit (Promega) according to the manufacturer’s instructions. The cells were transfected with pGL3 vector constructs and the pGL3-basic vector, along with the Renilla luciferase reporter vector pRL-TK (Promega) for normalization. The ratio of pGL3 to pRL-TK constructs was 10:1. After 48 h, all cells were rinsed with phosphate-buffered saline (Gibco) and analyzed using a dual-luciferase reporter assay system (Promega) according to the manufacturer’s instructions. The luciferase activity was measured using a Centro LB 960 luminometer (Berthold, Bad Wildbad, Germany). Promoter activity was analyzed using the ratio of firefly luciferase activity to Renilla luciferase activity.

### 2.7. Statistical Analysis

Data were obtained from three independent experiments and are presented as the mean ± standard error of the mean (SEM). All experimental results were statistically analyzed using the GraphPad Prism 9.5.1 software (GraphPad Software, La Jolla, CA, USA). An ordinary one-way analysis of variance (ANOVA) was used to assess the statistical significance. A *p*-value below 0.05 was deemed to be statistically significant (*, *p* < 0.05; **, *p* < 0.01; ***, *p* < 0.001).

## 3. Results

### 3.1. Identification of Genes with Porcine Endothelial Cell-Specific Expression

To develop promoters specific to porcine endothelial cells, we attempted to trace uniquely expressed genes in the pig vascular endothelium. Specific genes were identified by analyzing the transcriptome of GTKO PAECs in comparison with that of two other tissue-forming cell types—primary fibroblasts (GTKO PEFs) and a porcine renal epithelial cell line (pk-15).

RNA sequencing confirmed 8806 DEGs with a more than twofold difference in expression among the three cell types (Appendix A). We identified 243 transcripts as DEGs with over 10-fold higher expression in GTKO PAECs than in GTKO PEFs and pk-15 cells (Figure 1a). Next, these 243 genes were compared with genes specific to human endothelial cells, as indicated in the Human Protein Atlas “HPA; www.proteinatlas.org (accessed on 2 February 2023)”, which lists 195 enriched genes that are specific to human endothelial cells. Of the 243 genes in the GTKO PAECs, 34 were identified as being orthologous with the 195 genes listed in the HPA (Figure 1b). As shown in Figure 1c, we selected 10 genes that were expressed at substantially higher levels in GTKO PAECs—*ESAM*, *PEAR1*, protein tyrosine phosphatase receptor B (*PTPRB*), multimerin 2 (*MMRN2*), semaphorin 3G (*SEMA3G*), protocadherin 12 (*PCDH12*), platelet and endothelial cell adhesion molecule 1 (*PECAM1*), FYVE, RhoGEF and PH domain containing 5 (*FGD5*), SRY-Box transcription factor 18 (*SOX18*), and G protein subunit γ 11 (*GNG11*).

### 3.2. Potential of ESAM as an Endothelial Cell-Specific Promoter

In this study, we selected *ESAM* and *PEAR1* with over 500-fold higher expression in GTKO PAECs than in GTKO PEFs and pk-15 cells to study their feasibility for promoter development. Initially, we verified *ESAM* and *PEAR1* expression levels using qPCR with *ICAM2*, a well-known vascular endothelial cell-specific marker in pigs. The relative expression levels of *ESAM*, *PEAR1*, and *ICAM2* in the GTKO PAECs were significantly higher than those in the GTKO PEFs and pk-15 cells. *ESAM* showed the greatest relative fold change (approximately 1700-fold) in expression in the GTKO PAECs, followed by *PEAR1* (approximately 400-fold) and *ICAM2* (approximately 60-fold), indicating that the three genes were specifically expressed in the PAECs (Figure 2a). Next, we analyzed the relative expression levels of the three genes in the aorta and tissues of the major organs, including the heart, liver, lung, kidney, and spleen (Figure 2b). *ESAM* and *ICAM2* were strongly expressed in blood-vessel-rich lung tissue. *ESAM* expression was moderate in the aorta while being high in the heart and kidney. The expression pattern of *ESAM* was consistent with that observed in human tissues. *PEAR1* expression was non-specific in all examined tissues, including the vascularized lung tissue. In addition, *PEAR1* showed a markedly low level of expression in the aorta. Finally, we selected *ESAM* to develop endothelium-specific promoters since it was most significantly expressed in the GTKO PAECs and vascularized tissues.

### 3.3. Isolation of the Putative Promoter Region of Porcine ESAM

The porcine ESAM gene is located on chromosome 9, as is found in the mouse. The murine ESAM promoter, spanning 714 bp upstream of the translation start site, includes the binding site of a transcriptional factor, nuclear factor kappa B (NFκB), which has a crucial function in controlling transcription [28]. Figure 3a shows multiple putative NFκB-binding sites that are localized approximately 1.0 kb upstream of the porcine ESAM sequences. In addition, we observed that the number and size of the predicted CpG islands between porcine and murine ESAM promoter regions differed, showing higher G + C contents and an additional CpG island within 2 kb upstream in pigs (Figure 3b). Thus, we selected the 1.0 and 1.5 kb sequences containing two and three CpG islands that were located upstream of the translation start site of the porcine ESAM gene as the promoter regions ESAM1.0 and ESAM1.5, respectively.

### 3.4. Distinct Activity of the pESAM1.0 Promoter Leads to Specific Expression in Porcine Vascular Endothelial Cells

A dual-luciferase reporter system was applied to analyze the activity of the selected 1.0- and 1.5-kb length promoters, designated as ESAM1.0 and ESAM1.5, respectively. As depicted in Figure 4a, the ESAM1.0, ESAM1.5, and ICAM2 luciferase constructs were transiently transfected into PAECs, PEFs, and pk-15 cells. Notably, the ESAM1.0 promoter in PAECs led to the highest luciferase expression levels among the three promoters analyzed (*p* < 0.001), whereas the ESAM1.5 promoter exhibited lower luciferase expression than that of the ESAM1.0 promoter (Figure 4b). In GTKO PEFs, all three promoters showed comparable expression at low levels (Figure 4c). In pk-15 cells, the ESAM1.0 promoter demonstrated an approximately 4.6-fold greater level of promoter activity than that of the ICAM2 promoter (Figure 4d). These results indicate that ESAM1.0 promoter activity was more potent and had a comparable specificity to that of the well-known ICAM2 promoter.

## 4. Discussion

The xenotransplantation of pig organs is currently being tested in human clinical trials [29]. Recently, a 57-year-old man with end-stage heart failure was the first recipient in the world to receive a genetically modified porcine heart [30]. The establishment of transgenic pigs with profiles capable of regulating rejection mechanisms, such as thrombotic and immune responses, has led to remarkable advancements in the field of xenotransplantation [4]. Endothelial cells, which are the initial lineage communicating between the graft and host immunity, are the primary target of the host’s immune system after allogeneic transplantation [31]. Porcine endothelial cells in xenografted tissue induce the activation of the recipient’s immune system through antigen presentation, leading to the endothelial injury of donor tissue being recognized as one of the initial signs of rejection [14,32]. Immune-mediated reactions result in coagulation dysregulation within the blood vessels of the xenograft. Thus, targeting gene expression specifically in porcine endothelial cells is a strategic approach for improving transplant acceptance and decreasing immune-mediated complications.

Despite the critical function of porcine vascular endothelial cells in xenotransplantation studies, their transcriptomic properties have not been fully characterized. In this study, we attempted to identify the highly and specifically expressed genes in porcine vascular endothelial cells via RNA sequencing technology. We adopted endothelial cells and fibroblasts isolated from the aorta and ear skin, respectively, of α-Gal antigen-ablated GTKO pigs. GTKO is considered the minimum genetic background for preclinical xenotransplantation. Cells of the pk-15 epithelial cell line were also compared with endothelial cells, which comprise the primary structural and immunological barrier against foreign antigens. Transcriptomic comparative analysis identified 243 genes as highly and specifically expressed in the endothelial cells of GTKO pigs. To confirm which of these genes could be specific to endothelial cells, we performed further comparative analysis with an established and up-to-date database, HPA, which provides information regarding the localization and expression of proteins in human tissues and cells. We obtained 34 genes identified as endothelial cell-specific genes.

ESAM is one of the cell surface molecules that plays a role in immune cell transmigration at the vascular wall. *ESAM* ablation in the endothelial layer has been shown to disrupt immune cell development in mice [33]. The HPA database shows human ESAM to be highly expressed in blood-vessel-rich tissue, such as the lung. Figure 2b shows the presence of a higher level of *ESAM* expression in the lung and heart than in the other pig tissues examined. Eventually, we identified *ESAM* as the putative candidate with the highest potential for novel promoter development (Figure 1, Figure 2 and Figure 3). CpG islands are associated with mammalian gene promoters [34]. We predicted 2 and 3 CpG islands within 2.0 kb of the proximal sequences of murine and porcine *ESAM*, respectively. Given that the 714-bp-long murine *ESAM* promoter has NFκB-binding sites and putative CpG islands [28], we predict that the 1.0-kb-long porcine ESAM upstream sequence could play a role in promoter function (Figure 3b). We also selected a 1.5-kb-long upstream sequence with putative porcine-specific CpG islands to compare with the 1.0 kb sequence. Finally, we observed that the 1.0-kb-long ESAM1.0 sequence regulates the endothelial-specific expression of luciferase at a more significant level than that of the ICAM2 promoter and the 1.5-kb-long ESAM1.5 sequence. The ESAM1.5 promoter is distinguished by the presence of an additional third CpG island and NFκB-binding sites after the transcription initiation site. The methylation of CpG islands within the gene promoters has the potential to inhibit gene transcription [35,36]. The third CpG island from the translation initiation site in the ESAM1.5 promoter possibly conferred a repressive effect on transcription regulation. Taking these findings together, we were able to identify and develop the ESAM1.0 promoter, which led to high levels of expression of the gene, specifically in the porcine endothelial cells. These results suggest that the ESAM1.0 promoter is capable of regulating the expression of human anti-coagulant and anti-inflammatory genes at the endothelium and, thus, is useful when generating transgenic pigs.

The pig is considered to be a suitable human cardiovascular and metabolic disease model, owing to similarities in body size, anatomy, physiology, and, more importantly, a relatively extended lifespan. Furthermore, porcine endothelial cells could be extensively used as in vitro model systems in vascular biology and cardiovascular research, although vascular endothelial cell research in pigs is severely lacking compared with human and mouse studies. In this study, we provide a list of the DEGs in porcine endothelial cells revealed by transcriptome analysis and a comparison with the human database. We validated the expression patterns of only 3 genes of 34 human orthologues, *ICAM2*, *ESAM*, and *PEAR1,* in porcine cells and tissues. A comparative functional study of each protein translated from these 34 genes in the HPA still remains to be conducted. Nevertheless, the DEGs identified in this study will provide clues to generating not only transgenic pigs for xenotransplantation but also in vitro model systems for future vascular biology studies.

## 5. Conclusions

*ESAM*, a gene specific to porcine vascular endothelial cells, was identified through transcriptome analysis. *ESAM* expression was verified in the cells and tissues, and CpG islands and NFκB-binding sites aided in the prediction and design of promoters. We created the ESAM1.0 promoter, which was expressed specifically in porcine vascular endothelial cells, thereby allowing for precise control over gene expression in these cells, which is essential for vascular function and immune interactions in xenotransplantation.

## Figures and Tables

**Figure 1 genes-14-01928-f001:**
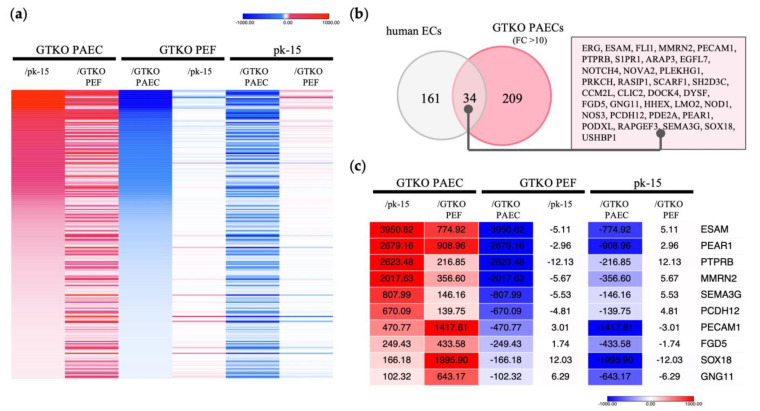
Identification of genes with endothelial cell-specific expression in GTKO pigs via RNA sequencing. (**a**) Heat map displaying strongly expressed genes specific to GTKO porcine aortic endothelial cells (PAECs). In comparing each cell type, red indicates a higher gene expression level, whereas blue indicates a lower level. A total of 243 transcripts were expressed 10-fold or more in GTKO PAECs, compared with GTKO porcine ear fibroblasts (PEFs) and pk-15 cells. (**b**) Comparison of genes specific to human endothelial cells (ECs). The gray circles depict 195 human EC-specific genes identified from the Human Protein Atlas “www.proteinatlas.org (accessed on 2 February 2023)”. The red circles represent 243 genes that presented with a 10-fold higher expression in GTKO PAECs. (**c**) Heat map displaying shared genes with significant expression differences. Genes with expression differences greater than 100-fold are shown among the 34 orthologues presented in (**b**). The numbers within each column represent the fold change in expression level.

**Figure 2 genes-14-01928-f002:**
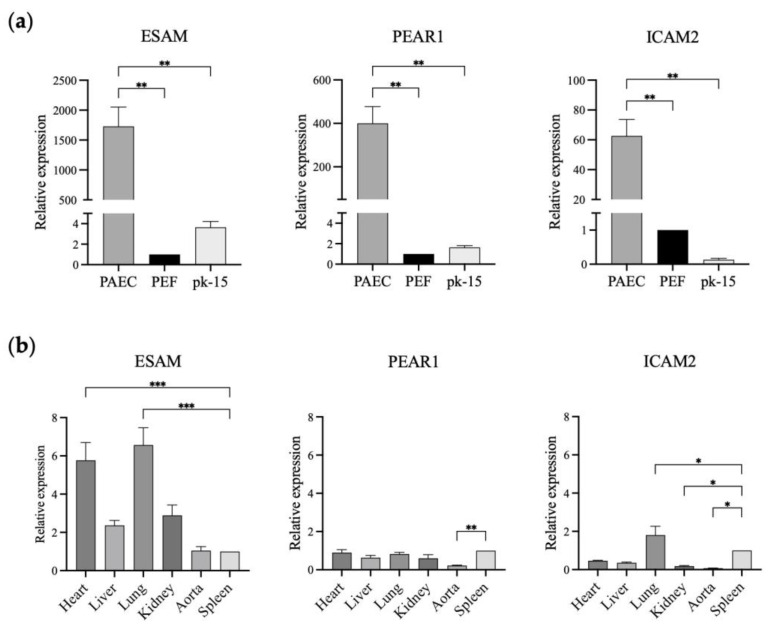
Relative expression of *ESAM*, *PEAR1*, and *ICAM2* in the different types of porcine cells and tissues. (**a**) *ESAM*, *PEAR1*, and *ICAM2* expression in GTKO porcine aortic endothelial cells (PAECs), GTKO porcine ear fibroblasts (PEFs), and pk-15 cells. (**b**) *ESAM*, *PEAR1*, and *ICAM2* gene expression in the tissues of wild-type miniature pigs. The relative expression of each gene was analyzed using PEFs (**a**) and spleen samples (**b**) as reference samples. *ESAM*, endothelial cell-specific adhesion molecule; *PEAR1*, platelet endothelium activation receptor 1; *ICAM2*, intercellular adhesion molecule 2. The results of experiments conducted in triplicate are expressed as the mean ± SEM. *, *p* < 0.05; **, *p* < 0.01; ***, *p* < 0.001.

**Figure 3 genes-14-01928-f003:**
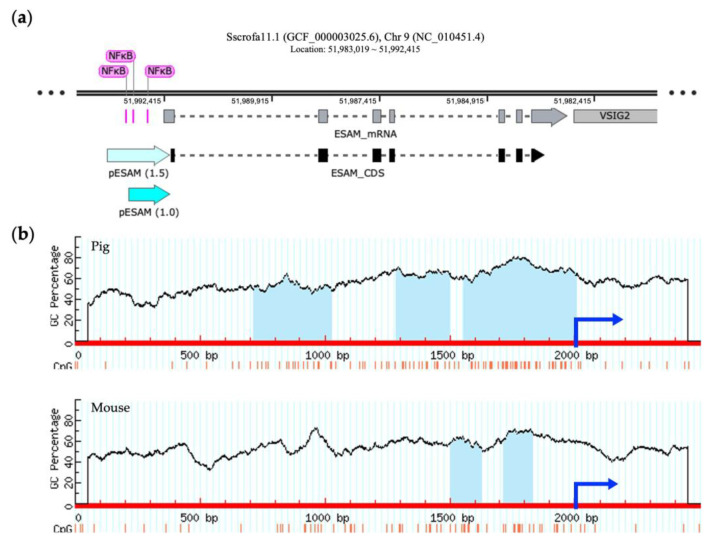
Identification of a potential promoter region in porcine ESAM. (**a**) Schematic representation of the structure of porcine *ESAM* using SnapGene software 7.0 (Dotmatics, Bishop’s Stortford, UK). The ESAM1.0 and ESAM1.5 promoter regions are denoted by light blue arrows. The segments colored pink represent the predicted NFκB-binding site. (**b**) Prediction of CpG islands in the 5′-flanking regions of porcine and murine *ESAM*. The red line represents the sequence of *ESAM*, the blue arrow represents the initiation site of the coding sequence (CDS) region, and the light blue area represents the predicted CpG island. NFκB, nuclear factor kappa B; *ESAM*, endothelial cell-specific adhesion molecule; *PEAR1*, platelet endothelium activation receptor 1; *ICAM2*, intercellular adhesion molecule 2.

**Figure 4 genes-14-01928-f004:**
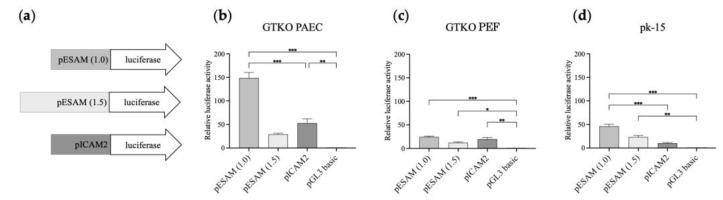
Comparative transcriptional activity of the ESAM1.0 and ESAM1.5 promoters with that of the ICAM2 promoter in different types of porcine cells. (**a**) Luciferase reporter vectors that were used to transfect (**b**) GTKO porcine aortic endothelial cells (PAECs), (**c**) porcine ear fibroblasts (PEFs), and (**d**) pk-15 cells. The pGL3 basic vector was used as a control. The results of experiments performed in triplicate are presented as the mean ± SEM. *, *p* < 0.05; **, *p* < 0.01; ***, *p* < 0.001. Luciferase pGL-3 vectors with different lengths of porcine ESAM promoter (1 and 1.5 kb) and porcine ICAM2 promoter were used for the construction of luciferase reporter vectors depicted in (**a**). *ESAM*, endothelial cell-specific adhesion molecule; *PEAR1*, platelet endothelium activation receptor 1; *ICAM2*, intercellular adhesion molecule 2.

**Table 1 genes-14-01928-t001:** Primers employed in this study.

Primer	Sequence (5′→3′)	
ACTB	(F)	CATCACCATCGGCAACGAGC
	(R)	TAGAGGTCCTTGCGGATGTC
18S rRNA	(F)	AAAGGAATTGACGGAAGGGC
	(R)	CCCACGGAATCGAGAAAGAG
ESAM	(F)	GCAAGGGGAGGTGTCTTCAA
	(R)	TGTGGCTCCACCGATGTATG
PEAR1	(F)	ATCCCCGAGAATGGCAACTG
	(R)	TTGCAGGGTACACAGCGTTT
ICAM2	(F)	ATCATCATCGCGGTCGTGTC
	(R)	CATGTAGGAACCTGTCCGCC

ACTB, actin β; 18S rRNA, 18S ribosomal RNA; ESAM, endothelial cell-specific adhesion molecule; PEAR1, platelet endothelium activation receptor 1; ICAM2, intercellular adhesion molecule 2.

## Data Availability

Data are contained within the article or the Appendix A.

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
