# Peer review of "Identification of the Porcine Vascular Endothelial Cell-Specific Promoter ESAM1.0 Using Transcriptome Analysis"

_genes, 2023, doi:10.3390/genes14101928_

Round 1
Reviewer 1 Report
This paper proposes an important research step in addressing xenograft rejection by developing a porcine endothelial cell-specific promoter. The research methodology is scientific, utilizing transcriptome analysis to screen 243 differentially expressed genes, further identifying the endothelial cell-specific adhesion molecule (ESAM) gene and validating it through qPCR. The establishment of the ESAM1.0 promoter provides a potential means for precise regulation of gene expression in porcine endothelial cells, which is crucial for vascular function and immune interactions in xenograft transplantation. However, there are some defects that need to be further improved.
1. In the discussion section, please cite previous research to discuss the possibility or evidence of the CPG island in the gene promoter region inhibiting gene transcription.
2. The fluorescence activity of the 1.5kb promoter sequence is significantly lower compared to the fluorescence activity of the 1.0kb promoter sequence. It is worth considering whether this difference is due to the presence of NFκB-binding sites. Is NFκB a transcription factor capable of inhibiting gene transcription? This question should be discussed and referenced in the relevant section of the discussion.
Author Response
Comments 1: This paper proposes an important research step in addressing xenograft rejection by developing a porcine endothelial cell-specific promoter. The research methodology is scientific, utilizing transcriptome analysis to screen 243 differentially expressed genes, further identifying the endothelial cell-specific adhesion molecule (ESAM) gene and validating it through qPCR. The establishment of the ESAM1.0 promoter provides a potential means for precise regulation of gene expression in porcine endothelial cells, which is crucial for vascular function and immune interactions in xenograft transplantation. However, there are some defects that need to be further improved.
Response 1: We are much obliged for giving the opportunity to submit a revised draft of our manuscript. We have revised our manuscript according to the reviewer’s suggested comments. Please see our point-by-point responses below for each comment.
Comments 2: In the discussion section, please cite previous research to discuss the possibility or evidence of the CPG island in the gene promoter region inhibiting gene transcription.
Response 2: As suggested by the reviewer, we cited previous studies in lines 293-294 and updated the citation in references.
Comments 3: The fluorescence activity of the 1.5kb promoter sequence is significantly lower compared to the fluorescence activity of the 1.0kb promoter sequence. It is worth considering whether this difference is due to the presence of NFκB-binding sites. Is NFκB a transcription factor capable of inhibiting gene transcription? This question should be discussed and referenced in the relevant section of the discussion.
Response 3: We would like to thank the reviewer for careful and thorough reading of this manuscript and for the constructive suggestions, which help to improve the quality of this manuscript. As illustrated in Figure 3, we presented that putative NFκB-binding sites are localized within third CpG islands sequences from the translation start site of the ESAM1.5 promoter. We assumed that methylation status of third CpG islands could interrupt activity of NFκB a transcription factor, consequently leading to repressive on luciferase expression. However, we cannot help hesitating to mention it in discussion section because of limited data and information. In this study, we aimed to identify a gene and its relevant sequences as a promoter to regulate higher expression with endothelial-cell specificity of the gene than that of ICAM2 promoter. The results were that ESAM1.0 promoter transcriptional activity was significant in PAECs, leading to 2.8-fold higher expression than that of the porcine ICAM2 promoter. We hope you are satisfied with our revision.

Reviewer 2 Report
Very important research and may help treatment use xenotransplantation
Author Response
Very important research and may help treatment use xenotransplantation
→ Thank you for your positive evaluation.

Reviewer 3 Report
- The methods section lacks an accurate explanation on the bioinformatic analysis of RNA-seq data. This section has to include information regarding the tool used for read mapping on the reference genome, data filtering and quality check, read quantification to obtain raw counts and sample normalization and gene expression quantification. In particular no information is available on the metric used to quantify gene expression levels from RNA-seq (CPM, TPM, RPKM, FPKM..)
- Moreover, there is no explanation on the statistical analysis used to identify differentially expressed genes. Even if not clearly stated, from the results shown in Figure 1 it seems that no replicates of the three cell lines were analyzed by transcriptome sequencing. If so, what kind of statistical analysis was employed? Please clarify this point, both regarding sample numbers and statistical analysis.
- In the method section, the Authors state that “filtered data were log2-transformed and subjected to quantile normalization”. In Figure 1a and 1c there are clearly gene expression comparisons in the linear scale, shown as relative fold changes with positive or negative values. Moreover, Figure 1A shows single sample comparisons, which makes the interpretation of gene expression levels and differences challenging and confounding. Please modify this representation by first turning gene expression quantifications in the log2 scale, then calculating the log2 ratio of each gene in each sample with respect to an average expression level for that gene in all samples. This way it will be possible to easily visualize the expression differences among all cell types. Change the legend and the scale bars accordingly.
- Supplementary figure S1 is lacking.
- Figure 2a could be shown in the log scale, so that the bars of the PEF and pk-15 samples could be seen.
- In Figure 2a and 2b there is no information on what is the reference sample, the one towards which the relative expression is calculated.
- In Figure 2a the Authors state that three replicates were performed. Actually the error bars are very high and the significance consequently lower than expected, considering the absolute expression differences shown. How do the Authors explain the variability in the expression levels of the genes in PAEC cells?
- There is no data regarding the number of pigs used to obtain PAEC and PEFs from GTKO pigs.
- The results lack data on transfection efficiency and eventual toxicity in the three lines of the vectors used for the dual-luciferase reporter assays. This is crucial to interpret reporter assay results.
- In the method section there is no reference to the quantification method used for real time data analysis (ddCt?).
- The analysis of gene expression in the aorta and tissues of the major organs is not clearly described in the methods section.
- Please include the name of the RNA extraction method used in paragraph 2.2.
- In the discussion section , lines 268-271 do not seem to be adequately supported by the results.
- RNA-sequencing raw data have to be deposited in a public repository such as SRA.
Author Response
We would like to thank the reviewer for careful and thorough reading of this manuscript and for the constructive suggestions, which help to improve the quality of this manuscript. We have addressed all issues indicated in the review report and showed in blue. We hope you are satisfied with our revision.
Comments 1: The methods section lacks an accurate explanation on the bioinformatic analysis of RNA-seq data. This section has to include information regarding the tool used for read mapping on the reference genome, data filtering and quality check, read quantification to obtain raw counts and sample normalization and gene expression quantification. In particular no information is available on the metric used to quantify gene expression levels from RNA-seq (CPM, TPM, RPKM, FPKM..)
Response 1: According to your suggestions, we updated 2.2. RNA sequencing in Materials and Methods (line 77∼94).
Comments 2: Moreover, there is no explanation on the statistical analysis used to identify differentially expressed genes. Even if not clearly stated, from the results shown in Figure 1 it seems that no replicates of the three cell lines were analyzed by transcriptome sequencing. If so, what kind of statistical analysis was employed? Please clarify this point, both regarding sample numbers and statistical analysis.
Response 2: As your suggestions, we added a new 2.3. DEG analysis section in Materials and Methods (line 95-110) to provide detailed information.
Comments 3: In the method section, the Authors state that “filtered data were log2-transformed and subjected to quantile normalization”. In Figure 1a and 1c there are clearly gene expression comparisons in the linear scale, shown as relative fold changes with positive or negative values. Moreover, Figure 1A shows single sample comparisons, which makes the interpretation of gene expression levels and differences challenging and confounding. Please modify this representation by first turning gene expression quantifications in the log2 scale, then calculating the log2 ratio of each gene in each sample with respect to an average expression level for that gene in all samples. This way it will be possible to easily visualize the expression differences among all cell types. Change the legend and the scale bars accordingly.
Response 3: Thank you very much for the thoughtful comments. In the Materials and Methods, 2.2. RNA sequencing has been thoroughly revised (line 77∼94).
Comments 4: Supplementary figure S1 is lacking.
Response 4: Thank you for pointing out what we missed. Supplementary figure S1 has been submitted as an attachment.
Comments 5: Figure 2a could be shown in the log scale, so that the bars of the PEF and pk-15 samples could be seen.
Response 5: According to your suggestions, we modified figure 2 (a)
Comments 6: In Figure 2a and 2b there is no information on what is the reference sample, the one towards which the relative expression is calculated.
Response 6: The reference sample is described in the legend for Figure 2 (line 186-187)
Comments 7: In Figure 2a the Authors state that three replicates were performed. Actually the error bars are very high and the significance consequently lower than expected, considering the absolute expression differences shown. How do the Authors explain the variability in the expression levels of the genes in PAEC cells?
Response 7: Thank you very much for the comments. In this study, we used two types of primary cells PAEC and PEFs isolated from two GTKO pigs, and commercial cell line pk-15 cells. It is known that primary cells are very heterogeneous on stage of senescence status. PEFs a type of primary cells were used as reference samples to obtain relative expression levels of the genes in PAEC. We presumed that comparison between highly heterogeneous cell types PAEC and PEFs would result the variability in the expression levels of the genes in PAEC cells. We hope you are satisfied with our revision.
Comments 8: There is no data regarding the number of pigs used to obtain PAEC and PEFs from GTKO pigs.
Response 8: We used primary cells isolated from a GTKO pig as described in previous studies (reference 19 and 20). Different passages of PAECs and PEFs were employed to analyze gene expression levels in the cell samples for qPCR and luciferase assays.
Comments 9: The results lack data on transfection efficiency and eventual toxicity in the three lines of the vectors used for the dual-luciferase reporter assays. This is crucial to interpret reporter assay results.
Response 9: We employed the Dual-Luciferase Reporter Assay System by Promega, a widely utilized commercial product, in accordance with the manufacturer's instructions. In the luciferase assay, cells were transfected with same amounts of vectors containing the ESAM1.0, ESAM1.5, and ICAM2 promoters. The pRL-TK vector constitutively expressing Renilla luciferase, provided by the manufacturer was co-transfected into cells to use an internal control reporter vector for normalization. We confirmed that the activity of Renilla luciferase did not vary significantly between samples.
Comments 10: In the method section there is no reference to the quantification method used for real time data analysis (ddCt?).
Response 10: We added new sentences to improve clarity in 2.4. Real-time quantitative PCR of the Materials & Methods (line 121-123)
Comments 11: The analysis of gene expression in the aorta and tissues of the major organs is not clearly described in the methods section.
Response 11: We modified 2.4. Real-time quantitative PCR section of the Materials & Methods (line 111-123)
Comments 12: Please include the name of the RNA extraction method used in paragraph 2.2.
Response 12: We extracted total RNA with the Trizol method and modified it (line 77 and 78).
Comments 13: In the discussion section , lines 268-271 do not seem to be adequately supported by the results.
Response 13: According to your suggestions, we deleted the sentence modified figure 2 (a)
Comments 14: RNA-sequencing raw data have to be deposited in a public repository such as SRA.
Response 14: Agree. We will submit the raw RNA sequencing data to a public database.

Round 2
Reviewer 3 Report
Thanks for addressing the raised issues.